# Violence against Women during the COVID-19 Pandemic in Mexico

**DOI:** 10.3390/healthcare11030419

**Published:** 2023-02-01

**Authors:** Leonor Rivera Rivera, Marina Séris Martínez, Luz Myriam Reynales Shigematsu, José Alberto Gómez García, Fernando Austria Corrales, Filiberto Toledano-Toledano, Alberto Jiménez Tapia, Diana Iris Tejadilla Orozco, Claudia I. Astudillo García

**Affiliations:** 1Centro de Investigación en Salud Poblacional, Instituto Nacional de Salud Pública (INSP), Cuernavaca 62100, Mexico; 2Secretariado Técnico del Consejo Nacional de Salud Mental (STCONSAME), Secretaría de Salud, Mexico City 06900, Mexico; 3Comisión Nacional para la Mejora Continua de la Educación (MEJOREDU), Mexico City 03900, Mexico; 4Unidad de Investigación en Medicina Basada en Evidencias, Hospital Infantil de México Federico Gómez, National Institute of Health, Mexico City 06720, Mexico; 5Unidad de Investigación Sociomédica, Intituto Nacional de Rehabilitación Luis Guillermo Ibarra Ibarra, Mexico City 14389, Mexico; 6Dirección de Investigación y Diseminación del Conocimiento, Instituto Nacional de Ciencias e Innovación para la Formación de Comunidad Científica, INDEHUS, Mexico City 14389, Mexico; 7Instituto Nacional de Psiquiatría Ramón de la Fuente Muñiz (INPRFM), Mexico City 14370, Mexico; 8Servicios de Atención Psiquiátrica (SAP), Secretaría de Salud, Mexico City 11410, Mexico

**Keywords:** violence against women, COVID-19, isolation, caregivers, binge drinking

## Abstract

This study measured the prevalence of cases of domestic violence against women and some associated factors during the COVID-19 pandemic in Mexico. Data were collected through a remote survey during 2020. The sample included 47,819 women aged 15 years and older. Jointpoint regression and logistic regression models were used. The prevalence of violence was 11.5%, which decreased in July and subsequently increased. The associated factors were being unemployed (OR = 2.01; 95%CI 1.89–2.16); being partially and totally quarantined (OR = 1.58; 95%CI 1.43–1.75 and OR = 1.47; 95%CI 1.32–1.63); being a caregiver of children; being a caregiver of elderly and/or suffering from a chronic illness (OR = 1.27; 95%CI 1.19–1.36; OR = 1.42; 95%CI 1.33–1.53; OR = 1.59; 95%CI 1.47–1.73); losing a family member to COVID-19 (OR = 1.26; 95%CI 1.13–1.41); and binge drinking (OR = 1.94; 95%CI 1.78–2.12). The confinement measures increased gender inequalities, economic problems and workload which further evidenced violence against women.

## 1. Introduction

Violence is based on a power imbalance in relationships, it involves the use of any kind of force as an attempt to undermine the will of others [1]. Violence has a major impact on the lives of millions of people globally. It is among the leading causes of death in the population between 15–44 years of age; it represents a considerable burden for health systems and, if prevented, its negative footprint may be significantly reduced [2].

Violence against women (VAW) is a problem that blocks the inclusive, equitable, and sustainable development of society; it is a global public health matter and a clear violation of human rights [3] affecting women’s physical, sexual, and mental health, as well as the social well-being of all the victims [4,5]. VAW also leads to direct and indirect violent deaths, causes morbidity for a multiplicity of health problems, and is linked to a number of risk behaviors [6]. Aggressors may exercise this type of violence in different environments or spaces, including the private and family space of the home [7], which does not necessarily represent a safe space for women [4]. Evidence shows that direct/indirect exposure to violence, and some contextual situations, such as sudden changes in economic status, can elicit aggressors to behave violently against women [8].

There are various factors associated with VAW that can be explained using the ecological model, which states that violence is the result of the interaction of individual, relational, community, and social elements [9]. Some of these factors have been identified as increasing the likelihood of VAW, such as age (younger women tend to suffer more violence), educational level (women with a lower education are more likely to become victims), and marital status (women who are not married to their partners are at greater risk) [8,10,11,12]. The factors related to VAW have been exacerbated by the COVID-19 pandemic, as the health, economic, and social crisis has generated stress and an increase in family arguments [2,12]. For most women, domestic work and caring for children, the sick, and elderly at home have also increased. This family burden has reduced women’s ability to avoid conflict with their aggressors, making them more vulnerable to psychological violence and sexual coercion [2].

When the COVID-19 pandemic began, international agencies, civil society organizations, and feminist groups warned that household VAW could increase [13,14]. The concern was generated by evidence that VAW often increases during crisis situations, as it did during the Ebola epidemic in 2014 [15]. The lockdown declared for most countries forced women to spend more time with their partners who could become abusive, making the study of VAW in these circumstances an important issue [16]. The evidence generated during the COVID-19 pandemic has confirmed an increase in the prevalence of VAW due to stressors and changes in daily routines [17,18]. A digital survey conducted in the U.S. found that 54% of women who were victims of violence before the pandemic continued to be victims, with an increase in physical and sexual violence at the onset of the lockdown [18]. A Mexican study during the pandemic confinement found a 6% prevalence of VAW, most of whom had already been victims of some type of violence prior to the pandemic [19]. This situation adds to the context of critical VAW in Mexico where 70% of women aged 15 or more have been victims of at least one violent event, and 43% of them experienced those events between 2020–21 during the pandemic, and 11% reported that the violence occurred at home [20].

Governments around the world have adopted policies to address VAW during the pandemic. Mexico implemented strategies to promote telephone helplines and specialized care centers, and a screening for violence was included in the survey “Remote Mental Health Care During the COVID-19 Pandemic” (“Atención Psicológica a Distancia para la Salud Mental por la contingencia por COVID-19”). However, analysis to date has not fully described the problem or other factors that have influenced its increase during this period. The objective of this study was to use the survey to determine the prevalence of and factors associated with household VAW during the COVID-19 pandemic in Mexico.

## 2. Materials and Methods

### 2.1. Study Design and Participants

We analyzed data from the survey, “Remote Mental Health Care During the COVID-19 Pandemic” [21], conducted from April to December 2020, involving 47,819 women aged 15 years and over who responded to the online questionnaire. The questionnaire was disseminated in the mental health section of the federal government’s coronavirus website, coronavirus.gob.mx, and in social media. The study was approved by the Research Ethics Committee of the Faculty of Psychology of the National Autonomous University of Mexico (FPSI/422/CEIP/157/2020).

### 2.2. Variables

The questions we used to measure the variables selected as violence-associated factors were designed based on relevant literature and the chosen theoretical framework of the subject.

#### Household Violence against Women

This variable was evaluated with the question: “Have you been a victim of any type of physical or verbal violence inside your home during the last month?”, with dichotomous response options: no = 0 and yes = 1.

### 2.3. Covariates

#### 2.3.1. Sociodemographic Characteristics

The categorical variables were age (15–19 years, 20–29 years, 30–39 years, 40–49 years, 50 years and over); marital status (single, married, widowed/divorced); and educational level (junior high school or less, high school, and undergraduate degree or more).

#### 2.3.2. COVID-19 Pandemic Variables

Variables related to the pandemic included employment status during the pandemic (employed or unemployed); isolation status by COVID-19, assessed with the question “Are you in isolation?”, with responses no = 0, partially (I must work or go out for food) = 1, and yes = 2; caregiver of children, determined with the question “Do you currently have children in your care?”, no = 0 and yes = 1; caregiver of an elder or of a person with chronic illness, assessed with two questions: 1) “Do you currently care for a person over 65 years of age?” and 2) “Do you currently care for a person with a chronic illness?”, where 0 = not a caregiver, 1 = caregiver of elder or person with chronic illness, and 2 = caregiver of elder and person with chronic illness; loss of family member due to COVID-19, with a question about loss in the past month, no = 0 and yes = 1; and excessive alcohol use, assessed by asking whether the respondent had consumed five or more alcoholic drinks in less than two hours during the past month, no = 0 and yes = 1.

### 2.4. Data Analysis

Frequencies and percentages were obtained for the variables of interest. The prevalence of VAW during the period of isolation was obtained and comparisons were made for sociodemographic variables using a chi-square test. A joinpoint regression analysis was performed, estimating the trend in the prevalence of violence in monthly time segments, in order to identify the point of significant change in the time series trend [22]. A logistic regression was also performed, adjusted for statistically and theoretically relevant variables, obtaining Odds Ratios (OR) and 95% Confidence Intervals (95% CI). The goodness-of-fit of the model was assessed using the Hosmer-Lemeshow test with Stata 15 (StataCorp. 2017).

## 3. Results

### 3.1. Sample Characteristics

Most of the women in the survey sample were 20–49 years old (76.5%). More than half reported a high level of education (undergraduate or graduate degree) and were single. Slightly more than a quarter reported being unemployed during the COVID-19 pandemic, and only 16% were not isolated. Almost 40% of the women reported having children and an elder or chronically ill person under their care, 6% lost a family member to COVID-19, and 9% reported excessive alcohol use in the month prior to the survey (Table 1).

### 3.2. Prevalence of Violence against Women

The prevalence of VAW was 11.5%, which decreased significantly to 7.8% in the month of July (β = −0.16, *p* < 0.01) and subsequently increased to a peak of 16.3% in the month of December (β = 0.30, *p* < 0.001), with an average monthly change of 15% (95% CI [9.9, 21.7], *p* < 0.01) (Figure 1). The prevalence of VAW was higher in those aged 15–19 years, those with a high school education, those who were unemployed or in total isolation (*p* < 0.001), caregivers of children, elders, or the ill, those who had lost a family member to COVID-19, and those who consumed alcohol excessively (Table 2).

### 3.3. Factors Associated with Household Violence against Women

The logistic regression model showed that younger women were more likely to experience violence than older women. Women who were unemployed during the COVID-19 pandemic were twice as likely to be victims of violence than those who were employed (OR = 2.01, 95% CI [1.89, 2.16]). Women who were partially or fully isolated were more likely to be victims of violence (OR = 1.58; 95% CI [1.43, 1.75] and OR = 1.47, 95% CI [1.32, 1.63], respectively). Regardless of the pandemic month in which the survey was completed, women were at risk of violence, and the risk increased from September onwards. Those who cared for children, or an elder or chronically ill person were more likely to be victims of violence, more so for those caring for elders or the chronically ill (OR = 1.59, 95% CI [1.47, 1.73]). Loss of a family member due to COVID-19 and excessive alcohol use were also associated with increased violence (OR = 1.26, 95% CI [1.13, 1.41] and OR = 1.94, 95% CI [1.78, 2.12], respectively) (Table 3).

## 4. Discussion

This study aimed at measuring the prevalence and some of the factors associated with household VAW during the COVID-19 pandemic in Mexico. Data were obtained from the survey “Remote Mental Health Care During the COVID-19 Pandemic” (“Atención Psicológica a Distancia para la Salud Mental por la contingencia por COVID-19”) which investigated mental health and addiction needs during the pandemic in Mexico [23]. The results showed that the prevalence of household VAW varied over the course of the pandemic and identified different factors associated with this violence, including unemployment, being a family caregiver, excessive alcohol use, and losing a family member to COVID-19.

The prevalence in our study (11.5%) is not comparable with the results reported by nationally representative surveys of Mexican women. However, it is similar to the results reported by one such survey conducted during the COVID-19 pandemic (11.4%) and as hypothesized, the data show an increase during the health emergency with respect to 2016 (10.3%) [20,24].

The results from other studies of VAW during the pandemic have been heterogeneous. Studies in Turkey and Iran reported higher prevalence of physical, sexual, and psychological VAW (27% to 40%) [25,26], and data from countries in Latin America, Africa, Asia, and the Middle East show that VAW increased during the pandemic [10]. Another study conducted in Mexico, however, found a prevalence of 6% [20]. These heterogeneous results could be related to the diverse ways of measuring VAW; different questions and temporalities, which makes it difficult to compare the prevalence of VAW in various settings. Yet, they show that the problem has a notorious presence in several developing countries, which combined with the precarious living conditions for most of the population in these countries, combines to create a very worrying public health scenario.

Our study found a U-shaped trend with important variations in the prevalence of VAW during the period of analysis. The prevalence was lowest in July 2020, and it peaked at 16% in December. This pattern could be explained using the “stress-aggression-remission” cycle of violence described by Walker [27], and it is probably linked to the increased use of alcohol during the December holidays, since a relationship has been found between the use of alcohol and the presence of violence [28,29].

As in previous studies, in the present investigation we found that younger women were more likely to be victims of violence during the COVID-19 pandemic [26,30], similar to the pre-pandemic situation. We also found that women who were married, or cohabiting, in a free union, separated, widowed, or divorced were more likely to experience violence than single women. This presents a worrisome scenario, given that public mental health policy in many countries is focused on recovery rather than prevention [10]; we believe that it is necessary to take advantage of the available evidence to implement preventive strategies aimed at women that can be applied in possible future pandemics.

Our results concur with those of studies prior to the pandemic that have reported the association between VAW and a non-married marital status [31,32,33], and evidence during the pandemic has also shown that marital status is an important risk factor for VAW [8]. Recent research conducted during the pandemic in Ethiopia found that women in arranged marriages were at higher risk of experiencing violence in their homes [30]. This evidence highlights the psychosocial complexity of romantic relationships in emergency contexts where there is likely to be an increase in emotional and relational tension.

Another finding of our study was that women who were unemployed during the pandemic were more likely to suffer violence. These results are consistent with those of other studies [21,33]. Some studies suggest that employment contributes to changing traditional gender roles, empowering women economically, which could contribute to a reducing dependence on their partners. Having paid employment could represent a protective factor against violence [32,34].

Women who were partially or fully isolated during the pandemic were more likely to experience violence. A study in India found that 24% of women reported increased intimate partner violence due to their inability to socialize, because they spent too much time at home, including working from home (22%) [35]. Other studies have shown that spending more time with the perpetrator at home can generate greater exposure to violent acts [36,37]. This suggests that staying at home not only has social implications, but also has an impact on women’s physical well-being and mental health [35].

Our results also showed that being a family caregiver during the pandemic was associated with an increased likelihood of becoming a victim of violence. This family burden could reduce women’s ability to avoid conflict within the household, leaving them more vulnerable to different types of violence [2]. There is evidence that women perceive that the increase in violence against them is related to the increased burden of household responsibilities [35]. These findings could represent the tip of an iceberg of gender inequality that still exists within families.

We also found that excessive alcohol consumption was associated with an increased risk of being a victim of domestic violence. Previous studies have documented that drug use, including alcohol use, is associated with the probability of VAW [38,39]. This association could be explained by the intoxication-violence model of Øverup et al. [28], in which alcohol use represents a strategy for coping with a difficult emotional situation, which could be associated with violent environments.

Our study has some limitations. Since it has a cross-sectional design, it is not possible to establish causality. Other aspects to consider is that our sampling strategy was non-probabilistic, and our survey was administered online, therefore our results cannot be extrapolated to all Mexican women. The measurement of household VAW was also made with a single question and without identifying the aggressor. Nevertheless, our results may be considered valid because they are consistent with other studies, and they provide information on the behavior of VAW during the most critical months of the COVID-19 pandemic.

## 5. Conclusions

Our results show that the prevalence of VAW during the COVID-19 emergency was higher than Mexico’s national data prior to the pandemic [24]. However, the way we investigated the presence of specific violence and the temporality of its occurrence (once in a lifetime vs. last month) prevent any valid comparison. However, our results are indicative of recent events that affected almost six thousand women during the outbreak. This situation, along with the other complex elements of a scenario of this nature, create an unflattering picture, because there is in Mexico a considerable number of women claiming they have suffered at least one form of violence [40].

Some other relevant aspects of our study are that it provides updated information on the behavior of VAW and that an immediate care service was offered to the women who participated and who needed the telepsychology service. However, it is necessary to complement this type of initiative with hotlines and other care services for male perpetrators of violence in order provide a more comprehensive response to VAW; by including the perpetrators, they are given the opportunity to work on their own problems with violence.

The COVID-19 pandemic triggered several factors that influenced VAW. Domestic work and the need to care for the children, the sick, the elderly, and other family members incremented during confinement; this, along with a boost in drug use as a coping strategy, the loss of jobs and the resulting financial limitations, the restriction in mobility, and the widespread insecurity, contributed in some way to encourage abusers to behave violently [28,41].

It is necessary to formulate contingency protocols to address this problem in case of health emergencies. Since unemployment and the burden of caregiving associated with VAW are results of gender inequality, it is also necessary to implement actions that socially and economically empower women by reducing unemployment, and such reforms as promoting microfinance and expansion of the labor market [41]. The results of this study provide baseline information on the phenomenon of VAW during the COVID-19 pandemic in Mexico. The VAW includes psychosocial factors that increase its complexity and calls for sociocultural, psychological, and sensitive interventions aiming at preventing it at a structural and situational level, focusing on the behavior of individuals, as well as the real ecological and contextual structure of the problem [8]. Further studies of this type are needed to measure the evolution of the phenomenon after the pandemic and to determine the impact of measures implemented to prevent it.

VAW could be reduced if more programs for prevention and work with gender, economic and social inequalities and inequities were reinforced and financed; in addition to modifying the norms and institutions that discriminate on the basis of gender and that contribute to fostering and perpetuating VAW. In Mexico there are governmental and civil society institutions that carry out prevention actions and provide care for women who suffer violence, but a reduction in VAW also requires re-education processes for men who perpetrate violence, for enabling them to generate new ways of expressing their masculinity.

## Figures and Tables

**Figure 1 healthcare-11-00419-f001:**
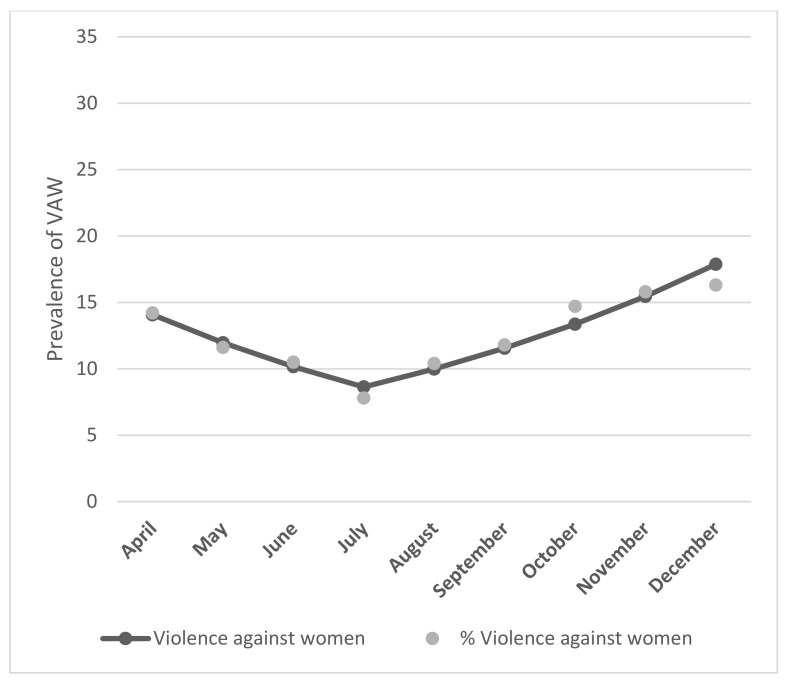
Violence against women during the COVID-19 pandemic: temporal analysis of prevalence with joinpoint regression. July: β = −0.16, *p* < 0.01, December: β = 0.30, *p* < 0.001, Average monthly change: 14.85%, 95% CI [9.9, 21.7], *p* < 0.01).

**Table 1 healthcare-11-00419-t001:** Sample Characteristics.

Sociodemographic Variables	*n* ^a^	%
Age group		
15–19	4848	10.1
20–29	12,762	26.7
30–39	14,086	29.5
40–49	9684	20.3
50 and older	6439	13.5
Educational level		
Junior high school or less	6021	12.6
High School	13,796	28.9
Undergraduate or graduate degree	28,002	58.6
Marital status		
Single	24,528	51.3
Married/domestic partner	18,534	38.8
Separated, widowed, divorced	4757	10.0
COVID-19 pandemic variables		
Employment status		
Unemployed	13,591	28.5
Employed	34,123	71.5
Isolation status		
No	7698	16.1
Partially	22,446	46.9
Yes	17,675	37.0
Caregiver of child		
No	29,501	61.7
Yes	18,318	38.3
Caregiver of elder and/or chronically ill person		
No	32,535	68.0
Caregiver of elder or chronically ill person	9199	19.3
Caregiver of elder and chronically ill person	6085	12.7
Loss of family member due to COVID-19		
No	44,874	93.8
Yes	2945	6.2
Excessive alcohol use in the past month ^b^		
No	43,225	91.2
Yes	4150	8.8

^a^ Total *n* = 47,819. ^b^ Consumption of five or more drinks in less than two hours in the past month.

**Table 2 healthcare-11-00419-t002:** Prevalence of Domestic Violence Against Women in the Past Month by Variables of Interest.

Prevalence in General Population	11.5%		
Sociodemographic Variables	*n* ^a^	%	*p*-Value
Age group			
15–19	767	15.8	<0.001
20–29	1585	12.4	
30–39	1450	10.3	
40–49	1007	10.4	
50 and older	687	10.7	
Educational level			
Junior high school or less	598	9.9	<0.001
High School	1660	12.0	
Undergraduate or graduate degree	3238	11.6	
Marital status			
Single	2774	11.3	0.086
Married/domestic partner	2131	11.5	
Separated, widowed, divorced	591	12.4	
COVID-19 pandemic variables			
Employment status			
Unemployed	2411	17.8	<0.001
Employed	3076	9.0	
Isolation status			
No	557	7.2	<0.001
Partially	2609	11.6	
Yes	2330	13.2	
Caregiver of child			
No	3212	10.9	<0.001
Yes	2284	12.5	
Caregiver of elder and/or chronically ill person			
No	3315	10.2	<0.001
Caregiver of elder or chronically ill person	1274	13.9	
Caregiver of elder and chronically ill person	907	14.9	
Loss of family member due to COVID-19			
No	5055	11.3	<0.001
Yes	441	15.0	
Excessive alcohol use in the past month ^b^			
No	4597	10.6	<0.001
Yes	806	19.4	

^a^ Total *n* = 47,819. ^b^ Consumption of five or more drinks in less than two hours in the past month.

**Table 3 healthcare-11-00419-t003:** Factors Associated with Domestic Violence Against Women During the COVID-19 Pandemic.

	OR (95% CI) ^a^	OR (95% CI) ^b^
Sociodemographic Variables		
Age group		
50 and older	1 (reference)	1 (reference)
15–19	1.57 (1.41–1.76) *	1.40 (1.22–1.60) *
20–29	1.19 (1.08–1.31) *	1.26 (1.13–1.41) *
30–39	0.96 (0.87–1.06)	1.02 (0.92–1.13)
40–49	0.97 (0.88–1.08)	1.00 (0.90–1.11)
Educational level		
High school or less	1 (reference)	1 (reference)
High School	1.24 (1.12–1.37) *	1.03 (0.93–1.15)
Undergraduate or graduate degree	1.19 (1.08–1.30) *	1.08 (0.98–1.20)
Marital status		
Single	1 (reference)	1 (reference)
Married/domestic partner	1.02 (0.96–1.08)	1.28 (1.19–1.38) *
Separated, widowed, divorced	1.11 (1.01–1.22)	1.47 (1.32–1.65) *
COVID-19 pandemic variables		
Employment status		
Employed	1 (reference)	1 (reference)
Unemployed	2.18 (2.06–2.31) *	2.01 (1.89–2.16)
Isolation status		
No	1 (reference)	1 (reference)
Partially	1.69 (1.53–1.86) *	1.58 (1.43–1.75) *
Yes	1.95 (1.77–2.15) *	1.47 (1.32–1.63) *
Month answered survey ^c^		
July	1 (reference)	1 (reference)
April	1.96 (1.76–2.19) *	1.61 (1.43–1.81) *
May	1.56 (1.40–1.74) *	1.32 (1.18–1.48) *
June	1.39 (1.25–1.55) *	1.21 (1.08–1.35) *
August	1.37 (1.22–1.55) *	1.29 (1.14–1.46) *
September	1.59 (1.40–1.80) *	1.48 (1.30–1.69) *
October	2.04 (1.80–2.31) *	1.76 (1.54–2.00) *
November	2.22 (1.93–2.57) *	1.77 (1.53–2.06) *
December	2.31 (1.91–2.81) *	1.93 (1.58–2.36) *
Caregiver of child		
No	1 (reference)	1 (reference)
Yes	1.17 (1.10–1.23) *	1.27 (1.19–1.36) *
Caregiver of elder and/or chronically ill person		
No	1 (reference)	1 (reference)
Caregiver of elder or chronically ill person	1.42 (1.32–1.52) *	1.42 (1.33–1.53)
Loss of family member due to COVID-19 ^d^		
No	1 (reference)	1 (reference)
Yes	1.39 (1.25–1.54) *	1.26 (1.13–1.41)
Excessive alcohol use in the past month		
No	1 (reference)	1 (reference)
Yes	2.03 (1.86–2.20) *	1.94 (1.78–2.12)

^a^ OR: Odds Ratio Unadjusted model (95% CI: 95% Confidence Interval). ^b^ OR: Odds Ratio Adjusted model (95% CI: 95% Confidence Interval); * *p* value < 0.05. ^c^ Month in which participants answered the online survey. ^d^ Loss of family member due to COVID-19 in the past month.

## Data Availability

The raw data supporting the conclusions of this article will be made available by the authors, without undue reservation.

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
