# Peer review of "Violence against Women during the COVID-19 Pandemic in Mexico"

_healthcare, 2023, doi:10.3390/healthcare11030419_

Round 1

Reviewer 1 Report

Dear authors,

Thank you for your submission. It is highly relevant as domestic violence and COVID-19 are of great concern worldwide and it is an asset to a journal such as healthcare.

I have no complaints of the content until table 1. Why do you have two rows for the table header?

I would not describe the population as young. Your sample has an adult population, as most people are between 20 and 49.

You must spell the disease as "COVID-19", not "Covid-19," just like AIDS, not Aids. Fix it in figure 1 caption.

In figure 1, bar chart with makes more sense because the data is discrete.

Consider using more charts to summarize the essential information, instead of large tables. You could send some of the information as supplementary material.

The manuscript is essentially well-written. Please work on the improvements.

Yours sincerely

Author Response

REVIEWER 1

REVIEWER COMMENT

AUTHORS’ RESPONSE

I have no complaints of the content until table 1. Why do you have two rows for the table header?

We appreciate the reviewer's comment. We have modified the table format. See Table 1.

I would not describe the population as young. Your sample has an adult population, as most people are between 20 and 49.

We thank the reviewer for his comment. We have modified the term “young” in almost all the manuscript.

You must spell the disease as "COVID-19", not "Covid-19," just like AIDS, not Aids. Fix it in figure 1 caption.

We appreciate the reviewer's comment. We have modified the term “Covid-19” for “COVID-19” along the manuscript.  See figure 1.

In figure 1, bar chart with makes more sense because the data is discrete.

We thank the reviewer for his comment. In this case we think the line chart better captures the idea of tracking violence changes over time, especially when small changes exist. See figure 1.

Consider using more charts to summarize the essential information, instead of large tables. You could send some of the information as supplementary material.

Thanks for the suggestion. We tried to summarize the information in charts, however, we consider it more pertinent to present our results in tables for a better understanding of our manuscript.

Reviewer 2 Report

The text is well written although Introduction and Conclusions should be improved.  In Introduction authors presented very narrow scientific background. It is restricted almost to the few main topics. I think that adding feminist perspective to this part of study will improve a lot the whole text. Additionally conclusions are very poor. If the Introduction will be improved, the authors will achieve new reasons for wider interpretation.

I suggest to rethink at least:

Nigam, S., & Soperna, N. (2017). A Study of the Understated Violence Within Social Contexts Against Adolescent Girls. Journal of Education Culture and Society, 8(2), 29–41. https://doi.org/10.15503/jecs20172.29.41

Seyyedyan, M., Moosazadeh , T. ., & Narimani Mostaali Begloo, M. . . (2021). Comparison of personality trait and emotional intelligence between the women who experienced domestic violence and control group. Journal of Education Culture and Society, 12(1), 212–223. https://doi.org/10.15503/jecs2021.1.212.223

Author Response

REVIEWER 2

REVIEWER COMMENT

AUTHORS’ RESPONSE

The text is well written although Introduction and Conclusions should be improved.  In Introduction authors presented very narrow scientific background. It is restricted almost to the few main topics. I think that adding feminist perspective to this part of study will improve a lot the whole text.

We thank the reviewer for his comment. We have included more background information to broaden the rationale. We also review the conclusions. However, we decided not to include a feminist frame because we worked based on the ecological model and we believe that feminist perspective requires a different approach to best analyze data.

Additionally conclusions are very poor. If the Introduction will be improved, the authors will achieve new reasons for wider interpretation.

We appreciate the reviewer's comment. We reviewed the conclusions and included more information and reflection about our findings.

I suggest to rethink at least:

Nigam, S., & Soperna, N. (2017). A Study of the Understated Violence Within Social Contexts Against Adolescent Girls. Journal of Education Culture and Society8(2), 29–41. https://doi.org/10.15503/jecs20172.29.41

Seyyedyan, M., Moosazadeh , T. ., & Narimani Mostaali Begloo, M. . . (2021). Comparison of personality trait and emotional intelligence between the women who experienced domestic violence and control group. Journal of Education Culture and Society12(1), 212–223. https://doi.org/10.15503/jecs2021.1.212.223

Thanks for the suggestion. We reviewed the proposed references and included some rationale and data that we considered useful.

Reviewer 3 Report

Introduction does not provide a very strong rationale for the study

Methods-

1.       The authors should mention if any kind of dissemination exercise was carried out to inform women regarding the online survey. Is it possible that only a certain section of the population might have been aware of the survey and answered it?

2.       On what basis were the questions used to identify factors associated with violence included in the survey?

Results:

1.       The terms “RM” is used in Section 3.3. Please provide expansion when it first appears in the manuscript.

2.       Table 2- should give absolute numbers of women in each category who have reported to have experienced violence. At present it provides only percentages

3.       Tables 2 and 3

a.       Should be combined to present results of unadjusted and adjusted regression analysis in the same table which is the standard way of presenting such results

b.       The columns can be factors, number of women with violence and %, Unadjusted OR with 95% CI, adjusted OR with 95% CI for factors that went into the final model

Discussion: The following points can be covered in discussion

1.       Is the sample characteristics comparable to general female population composition of Mexico based on any national level estimates?

2.       Are the findings from the levels of violence known to expect during non-pandemic periods?

3.       What additional information has the study provided and what is the implication of these findings?

Discussion: The recommendations provided in conclusion seem very generic and have to be implemented to curb violence even in the absence of pandemic

Author Response

REVIEWER 3

REVIEWER COMMENT

AUTHORS’ RESPONSE

Introduction

Introduction does not provide a very strong rationale for the study

We appreciate the reviewer's comment. We have included more background information to strengthen the rationale of the manuscript.

Methods

 The authors should mention if any kind of dissemination exercise was carried out to inform women regarding the online survey. Is it possible that only a certain section of the population might have been aware of the survey and answered it?

We thank the reviewer for his comment. As indicated in the method section, the analytical sample is derived from the ""Remote Mental Health Care During the COVID-19 Pandemic” survey, which was addressed to the general population and disseminated on social media, and in the mental health section of the federal government’s coronavirus website, coronavirus.gob.mx, which the mexican government enabled to report on the coronavirus situation. Given this, the data can only be extrapolated to those who found out about the survey and had access to electronic means to answer it, which is indicated as a limitation in the discussion section.

 On what basis were the questions used to identify factors associated with violence included in the survey?

We appreciate the reviewer's comment. The creation of the questions used to define the risk factors associated with violence against women was based on a literature review and on the theoretical framework linked to this type of violence. We included the following text to state it in the manuscript: “The questions we used to measure the variables selected as violence associated factors were designed based on what literature and the selected theoretical framework on the subject report as relevant.” See page 3.

Results

The terms “RM” is used in Section 3.3. Please provide expansion when it first appears in the manuscript.

We thank the reviewer for his observation. We have modified the term “RM” for “OR”.  See section 3.3.

Table 2. should give absolute numbers of women in each category who have reported to have experienced violence. At present it provides only percentages

We appreciate the reviewer's comment. We have modified the Table 2 according to the comment. See table 2.

a.  Table 2 and 3 should be combined to present results of unadjusted and adjusted regression analysis in the same table which is the standard way of presenting such results.

b.       The columns can be factors, number of women with violence and %, Unadjusted OR with 95% CI, adjusted OR with 95% CI for factors that went into the final model

Thanks for the suggestion. We tried to combined the table 2 and 3, nevertheless, we consider it more pertinent to present our results in two tables for a better understanding of the results of our manuscript.

In the Table 3 we added the unadjusted OR with 95%CI. See table 3.

Discussion

The sample characteristics comparable to general female population composition of Mexico based on any national level estimates?

We appreciate the reviewer's comment. We added some information regarding national data in the first paragraph of page 8. 

Are the findings from the levels of violence known to expect during non-pandemic periods?

We thank the reviewer for his insightful inquiry. We have included some information to answer the question. See page 8, paragraph 2. 

What additional information has the study provided and what is the implication of these findings?

We appreciate the reviewer's comment. We included more information to respond to this question in the discussion. 

Discussion: The recommendations provided in conclusion seem very generic and have to be implemented to curb violence even in the absence of pandemic

Thanks for the suggestion. We have included some modifications in our conclusions to make them more specific.

Round 2

Reviewer 3 Report

Thanks for taking the effort to edit the manuscript.

1.       Few of my concerns remain unaddressed.

1.1. Recommendations added in the conclusion remain generic, and are measures that have been known to be important to prevent VAW.

1.2. I had requested for a comparison of the population of this study with that of national level estimates. The reason being that the study has women predominantly in the 20-49 age group. This is expected considering this was an online survey. It would have been good to know how different the study population is from the national population.

2.       I understand now that these are some issues which cannot be corrected at this stage as the study is completed now. Though the questionnaire has been reported to have developed based on review of literature (mentioned as “what literature” in text) and theories around VAW (not made explicit), some key variables which could be of relevance to the pandemic are missed. These could have been captured on an online survey. For example, access to support services, changes in financial status due to COVID-19, prior history of VAW, variables linked to partners

3.       The revised version contains some statements which do not seem to convey any concrete meaning, or perhaps beyond my understanding of the language. For example-

3.1. Line 217-220- “Although information on prevalence and trends of VAW is not scarce, there is a lack of evidence regarding prevention and services for survivors; many countries are more focused on recovery and little has been done to review policies and programs that consider the lessons that we should have learned for upcoming pandemics.” What is the relevance in this manuscript? This study has not addressed this gap.

3.2. Line 273-274- “However, it is necessary to complement this type of initiative with hotlines for male perpetrators of violence.”- What are these hotlines for male perpetrators expected to do?

3.3. Line 294- “…the reduction of VAW also requires re-education processes for men who perpetrate violence which may generate new ways of living masculinity”- What is meant by “Living masculinity”

In conclusion, I am not convinced about accepting this manuscript with its current shortcomings. I shall defer to the editors for their final decision in this matter.

Author Response

REVIEWER 3

REVIEWER COMMENT

AUTHORS’ RESPONSE

1.     Few of my concerns remain unaddressed.

2.      

3.     Recommendations added in the conclusion remain generic, and are measures that have been known to be important to prevent VAW.

We appreciate the reviewer's comment. We made new adjustments to our conclusion. See page 10.

4.     I had requested for a comparison of the population of this study with that of national level estimates. The reason being that the study has women predominantly in the 20-49 age group. This is expected considering this was an online survey. It would have been good to know how different the study population is from the national population.

Thank you very much for your comment. The majority of our study population was between 20 and 49 years (76.5%). This percentage is slightly higher than the one reported in the mexican women population with the same age range (62.2%). This is understandable since it was a convenience sample and not a probability sample. We mentioned this point in the limitations of our paper. See page 9, lines 253-255.

5.     I understand now that these are some issues which cannot be corrected at this stage as the study is completed now. Though the questionnaire has been reported to have developed based on review of literature (mentioned as “what literature” in text) and theories around VAW (not made explicit), some key variables which could be of relevance to the pandemic are missed. These could have been captured on an online survey. For example, access to support services, changes in financial status due to COVID-19, prior history of VAW, variables linked to partners

We appreciate the reviewer's comment. As you commented, we had limitations in our study which cannot be amended at this point. However, we consider your comment very pertinent and our team  will take it into account for future studies.

6.     The revised version contains some statements which do not seem to convey any concrete meaning, or perhaps beyond my understanding of the language. For example-

 Line 217-220- “Although information on prevalence and trends of VAW is not scarce, there is a lack of evidence regarding prevention and services for survivors; many countries are more focused on recovery and little has been done to review policies and programs that consider the lessons that we should have learned for upcoming pandemics.” What is the relevance in this manuscript? This study has not addressed this gap.

Line 273-274- “However, it is necessary to complement this type of initiative with hotlines for male perpetrators of violence.”- What are these hotlines for male perpetrators expected to do?

 Line 294- “…the reduction of VAW also requires re-education processes for men who perpetrate violence which may generate new ways of living masculinity”- What is meant by “Living masculinity”

In conclusion, I am not convinced about accepting this manuscript with its current shortcomings. I shall defer to the editors for their final decision in this matter.

We thank the reviewer for the thorough revision. We modified  these unclear statements to try to make them clearer.

See page 8 in lines 214-218

This presents a worrisome scenario, given that public mental health policy in many countries is focused on recovery rather than prevention [10]; we believe that it is necessary to take advantage of the available evidence to implement preventive strategies aimed at women that can be applied in possible future pandemics.

See page 9 in lines 271-274

However, it is necessary to complement this type of initiative with hotlines and other care services for male perpetrators of violence in order provide a more comprehensive response to VAW; by including the perpetrators, they are given the opportunity to work on their own problems with violence.

See page 10 in lines 297-301.

…the reduction of VAW also requires re-education processes for men who perpetrate violence which may generate new ways of expressing masculinity,